# A Novel Mitovirus PsMV2 Facilitates the Virulence of Wheat Stripe Rust Fungus

**DOI:** 10.3390/v15061265

**Published:** 2023-05-28

**Authors:** Yanhui Zhang, Hualong Guo, Siyu Zhou, Daipeng Chen, Gang Xu, Zhensheng Kang, Li Zheng

**Affiliations:** 1State Key Laboratory of Crop Stress Biology for Arid Areas, College of Plant Protection, Northwest A&F University, Yangling 712100, China; zhangyanhui0323@163.com (Y.Z.); guohualong01004@163.com (H.G.); 2Hainan Yazhou Bay Seed Laboratory, Sanya Nanfan Research Institute of Hainan University, Sanya 572025, China

**Keywords:** *Puccinia striiformis*, mycovirus, mitovirus, PsMV2, pathogenicity-enhancing

## Abstract

Wheat stripe rust, caused by the obligate biotrophic fungus *Puccinia striiformis* f. sp. *tritici* (*Pst*), seriously affects wheat production. Here, we report the complete genome sequence and biological characterization of a new mitovirus from *P. striiformis* strain GS-1, which was designated as “Puccinia striiformis mitovirus 2” (PsMV2). Genome sequence analysis showed that PsMV2 is 2658 nt in length with an AU-rich of 52.3% and comprises a single ORF of 2348 nt encoding an RNA-dependent RNA polymerase (RdRp). Phylogenetic analysis indicated that PsMV2 is a new member of the genus *Unuamitovirus* within the family *Mitoviridae.* In addition, PsMV2 multiplied highly during *Pst* infection and it suppresses programmed cell death (PCD) triggered by Bax. Silencing of PsMV2 in *Pst* by barley stripe mosaic virus (BSMV)-mediated Host Induced Gene Silencing (HIGS) reduced fungal growth and decreased pathogenicity of *Pst*. These results indicate PsMV2 promotes host pathogenicity in *Pst*. Interestingly, PsMV2 was detected among a wide range of field isolates of *Pst* and may have coevolved with *Pst* in earlier times. Taken together, our results characterized a novel mitovirus PsMV2 in wheat stripe rust fungus, which promotes the virulence of its fungal host and wide distribution in *Pst* which may offer new strategies for disease control.

## 1. Introduction

Mycoviruses (or fungal viruses) are widely present in all major taxa of plant pathogenic fungi [1,2], and persistently infect the host, which may play an important ecological role in nature [3]. Mycoviruses usually contain genomes of double-stranded RNA (dsRNA), positive-sense single-stranded RNA (+ssRNA), negative-sense ssRNA (-ssRNA), rare ssDNA circular genomes, and RT-ssRNA genomes (https://talk.ictvonline.org/, accessed on 1 December 2022) [4]. Recently, with next-generation sequencing (NGS) development, an increasing number of mycoviruses have recently been characterized [5]. The discovery of novel mycoviruses expands our knowledge of virus taxonomy, ecology, and evolution [6,7]. Although a large number of mycoviruses are cryptic or asymptomatic, several mycoviruses causing apparent phenotypic alterations, including attenuation of fungal virulence (hypovirulence), is of particular interest for biocontrol purposes [7,8,9]. Thus far, the +ssRNA mycoviruses identified are mainly classified into nine families (*Alphaflexiviridae, Barnaviridae, Botourmiaviridae, Deltafleiviridae, Endornaviridae, Gammaflexiviridae, Hypoviridae, Mitoviridae,* and *Narnaviridae*) (https://talk.ictvonline.org/taxonomy/, accessed on 1 December 2022) [10]. The best-known example of virus-mediated hypovirulence was that Cryphonectria hypovirus 1 (CHV1) had been successfully used for the biological control of a devastating chestnut disease in Europe [9]. In addition, other mycovirus infections in pathogenic fungi have also been identified as promising resources for biological control [11,12,13].

The genus *Mitovirus* has been assigned to the newly established family *Mitoviridae.* Recent studies support that *Mitovirdae* is divided into at least four new genera: *Unuamitovirus, Duamitovirus, Triamitovirus,* and *Kvaramtovirus* (https://talk.ictvonline.org/files/proposals/taxonomy_proposals_fungal1/m/fung04/13098, accessed on 12 November 2022). Viruses in the family *Mitoviridae* have non-encapsidated +ssRNA virus genomes, do not form any infectious virions and replicate in the mitochondria of host cells [14,15]. A typical mitovirus genome is 2.0 to 4.5 kb in length and contains a single open reading frame (ORF) that encodes an RNA-dependent RNA polymerase (RdRp) and has six conserved amino acid motifs (I–IV) [16,17]. The genome of mitoviruses with typical characteristics: 5′ and 3′ untranslated regions (UTRs) of varying sizes [18]; using UGA codons to encode a tryptophan (Trp) rather than stop codon for translation termination; prefer to use stop codons of UAA and sometimes UAG [19,20]; the terminal sequences could fold into stable secondary genome structures which play a key role in replicating and genomes stability [21,22]. Mitoviruses are widespread among phytopathogenic fungi [23,24,25]. While some mitoviruses can lead to the hypovirulence of their fungal hosts [26], others have beneficial effects on virulence. For example, the Fusarium circinatum mitovirus 1 (FcMV1) can promote the pathogenicity of its host [27]. Mitoviruses may also provide key insights into the fundamental biology of virus-host interactions. Mitovirus-related sequences identified in plants can be integrated into the nuclear and mitochondrial genomes as DNA copies [26]. In Arabidopsis, such integration of mitovirus-related sequences may function as part of an RNA-mediated defense system [28].

Wheat stripe rust (yellow rust), caused by *Puccinia striiformis* f. sp. *tritici* (*Pst*), is a worldwide airborne fungal disease that threatens the safety of wheat production [29,30,31]. Compared with the study of viruses in other fungi, an in-depth study of mycoviruses in rust fungus is limited. So far, seven novel mycoviruses have been identified from *Pst*, including *Mitovirus, Narnavirus*, and *Totivirus*, but their potential effects on their hosts still need to be investigated [32,33,34,35]. In this study, we report a novel mitovirus, Puccinia striiformis mitovirus 2 (PsMV2) from *Pst*, and ascertain its taxonomic status. Furthermore, we characterized the biological effect of mitovirus PsMV2 in wheat stripe rust, and PsMV2 plays a positive role in *Pst* virulence. This study not only expands our understanding of the origin, ecology, and evolution of mycoviruses, but also provides insight into the interactions among mycoviruses, fungal phytopathogens, and plant hosts.

## 2. Materials and Methods

### 2.1. Fungal Strains, Plant Materials and Growth Conditions

*Pst* strain GS-1 (race CYR31) was isolated from wheat stripe rust fungus in Gansu province, China, in 2021. Firstly, the *Pst* strain GS-1 was maintained and propagated on the susceptible wheat variety Mingxian 169 as previously reported [33]. Fresh urediospores were collected from infected plant leaves and stored at −80 °C for RNA or DNA extraction until further use [36]. Wheat cultivar Su11 was grown in a controlled environment chamber under 8/16 h night/day conditions at 16 °C, which is highly susceptible to *Pst* strain GS-1 (race CYR31). Then, Su11 was used for the analysis of gene transcription levels and barley stripe mosaic virus (BSMV)-mediated host-induced gene silencing (HIGS) assays. Wheat seedlings were cultivated, inoculated with *Pst*, and maintained according to procedures and conditions as previously described [37].

### 2.2. Extraction and Purification of dsRNAs

The dsRNA was obtained from 1 g of *Pst* urediospores by selective absorption to a cellulose powder CF-11 (Whatman, UK) column containing 16% ethanol [32]. The dsRNA extraction was carried out according to the previously published protocol [38]. The obtained dsRNAs were further treated with DNase I and S1 nuclease (TaKaRa, Dalian, China) to digest genomic DNA and ssRNA (TaKaRa), the qualities of which were then analyzed based on 1.0% (*w*/*v*) agarose gel electrophoresis.

### 2.3. Total RNA Extraction, cDNA Cloning and Sequencing

Total RNA was extracted from *Pst* strain GS-1 urediospores. The extracted total RNA was further treated with RQ1 RNase-Free DNase (Promega Corp, Madison, WI, USA) to eliminate fungal DNA. Cloning, sequencing, and analyzing of the cDNAs for the dsRNAs by random primer-mediated PCR were conducted as previously described [38,39,40]. The total RNA was also used for reverse transcription using the RevertAid First Strand cDNA Synthesis Kit for qPCR (Thermo, MA, USA). PCR amplification was performed using Taq PCR Master Mix (CWBIO, Beijing, China). Each purified segment was ligated to the pMD18-T vector (Takara, Dalian, China) for Sanger sequencing. The terminal sequences of PsMV2 were cloned by rapid amplification of ligase-mediated rapid amplification of cDNA ends (RLM-RACE). The PCR primers and their sequences mentioned above are listed in Appendix A. The full-length cDNA sequence was obtained by assembling the partial sequences in different cDNA clones. In both orientations, each base was determined with at least five independent overlapping clones. A genome of length 2658 nt was obtained from urediospores clones derived from *Pst* strain GS-1 (race CYR31).

### 2.4. Sequence and Phylogenetic Analysis

Open reading frames (ORFs) and conserved domain(s) were determined using ORF Finder program and CD-search in the National Center for Biotechnology Information (NCBI) (http://www.ncbi.nlm.nih.gov, accessed on 3 June 2022), and predictions were made by selecting the “standard” genetic code for all viral contigs. Motifs searches were conducted in the PROSITE database (http://www.expasy.ch/, accessed on 12 October 2022). Protein molecular mass was predicted using the Protein Molecular Weight tool (https://www.bioinformatics.org/sms/prot_mw.html, accessed on 12 October 2022). Potential secondary structures at the terminal sequences of the viral genome sequence were predicted using Mfold RNA structure software [41]. DNAMAN 7.0 and ClustalW 2.0 were used for multiple alignments of nucleotide and coding amino acid sequences [42]. The phylogenetic tree was constructed using the Maximum likelihood (ML) method and the Jones–Taylor–Thornton (JTT) model of the Molecular Evolutionary Genetics Analysis (MEGA) software version 11.0 programs with 1000 bootstrap replicates. See Appendix A for all virus names and viral protein accession numbers used for phylogenetic analysis.

### 2.5. RT-PCR Analysis

To assay transcript levels of PsMV2, urediospores and leaves of wheat Su11 infected with urediospores of CYR31 (strain GS-1) were harvested at 6, 18, 24, 48, 72, 96, 120, 148, 196, 216 or 256 h post-inoculation (hpi). To verify the host range of the PsMV2, we performed RT-PCR for a range of plant and fungal samples using a specific primer pair (PsMV2-RdRp-F/R). RNA was extracted from all samples using the Quick RNA Isolation Kit (Huayueyang Biotechnology, Beijing, China). For the cDNA synthesis, 3 μg of RNA was used by the RevertAid First Strand cDNA Synthesis Kit (Thermo, MA, USA). The PCR amplifications were carried out with 2 μL of the synthesized cDNA and the Taq PCR Master Mix (CWBIO, Beijing, China). The products of the RT-PCR amplifications were analyzed by electrophoresis in 1.5% agarose gels. qRT-PCR was performed on a CFX Connect Real-Time System (BioRad, Hercules, CA, USA) with a 20 μL reaction mixture containing 10 μL ChamQ SYBR qPCR Master Mix (Vazyme, Nanjing, China), 2 μL diluted cDNA (1:5), 7 μL distilled H_2_O, 0.5 μL forward primer (10 μM) and 0.5 μL reverse primer (10 μM). The primers are listed in Appendix A. *PstEF*1 and *TaEF*1 were chosen as the internal control genes to normalize the RNA level of *Pst* and wheat leaves, respectively. Raw data were converted into expression data by the 2^−ΔΔCt^ method. Each sample was analyzed as three biological replicates and each PCR analysis included three technical replicates.

### 2.6. Agrobacterium Tumefaciens Infiltration Assays

To detect the inhibitory activity of PsMV2 on Bax-induced cell death, *A. tumefaciens*-mediated transient expression method was used. The RdRp nucleotide sequence of PsMV2 was amplified using specific primers PsMV2-PVX-RdRp-F/R (Appendix A). Coding sequences of PsMV2 were subsequently ligated with potato virus X (PVX) vector pGR106 (for transient expression in *N. benthamiana*) to construct the PsMV2-GFP plasmid, using ClonExpress II One-Step Cloning Kit (Vazyme, Nanjing, China). This plasmid was separately transformed into *A. tumefaciens* GV3101 (WEIDI, Shanghai, China), which was cultured in Luria–Bertani medium with 20 mg/L rifampicin and 50 mg/L kanamycin. The infiltration for the tobacco method was described previously [43]. GFP served as a negative control, and then *PstGSRE1* can suppress proapoptotic protein Bax-induced PCD used as a positive control. Symptoms were monitored and recorded from 5 to 6 days after infiltration. Each experiment was carried out using three independent biological replicates.

### 2.7. BSMV-Mediated Silencing

Two cDNA fragments of PsMV2 were selected for barley stripe mosaic virus (BSMV)-mediated Host Induced Gene Silencing (HIGS). To make sure the specificity for PsMV2 silencing, PsMV2-1as and PsMV2-2as were analyzed by BLASTN. The BSMV: PsMV2-1as and BSMV: PsMV2-2as were cloned and inserted into BSMV as previously described [44]. Capped in vitro transcripts of BMSV genomes with BSMV: γ: PsMV2-1as, BSMV: γ: PsMV2-2as, BSMV: γ: TaPDS-as, BSMV: γ, BSMV: α, and BSMV: β were generated in vitro with the RiboMAX Large-Scale RNA Production System-T7 and the Ribom7G Cap Analog (both by Promega) according to the manufacturer’s instruction. The BSMV RNA genomes BSMV: α, BSMV: β with BSMV: γ, or γ-TaPDS-as, or BSMV: γ: PsMV2-1as, or BSMV: γ: PsMV2-2as were mixed in a 1:1:1 ratio and mechanically inoculated onto the second leaf of the two-leaf-stage wheat with 1× FES buffer (0.5:0.5:0.5:8.5) by gentle rubbing of the surface with a gloved finger. BSMV: TaPDS-as was used as the positive control. Mock-treated wheat seedlings inoculated with only 1× FES buffer (Sodium pyrophosphate 0.5 g, Bentonite 0.5 g, Kieselguhr 0.5 g, Dipotassium phosphate 2.613 g, Glycine 1.877 g, ddH_2_O to 50 mL) or BSMV: γ were served as a negative control. All BSMV-inoculated wheat seedlings were retained in controlled environmental chambers at 25 ± 2 °C with 16 h of light and 8 h of darkness. At 12 days after virus inoculation, the leaf symptoms were examined. When photobleaching symptoms appeared on leaves silenced with BSMV: TaPDS, the fourth leaves of seedlings were inoculated with fresh *Pst* race CYR31 (strain GS-1) urediospores on Suwon11 and maintained at 16 ± 2 °C. To evaluate the silencing efficiency and biomass ratio, *Pst*-infected leaves were analyzed using RT-qPCR assay. Phenotypes of wheat inoculated with *Pst* race CYR31 (strain GS-1) were photographed 14 days after inoculation. Three independent biological replicates were performed for each experiment.

### 2.8. Histological Observation of Pst Growth in Wheat Leaves

The wheat leaves inoculated with BSMV were collected at 24, 48, and 120 hpi for histochemical analysis as previously described [45]. Stained leaf segments were fixed and cleared in ethanol/acetic acid (1:1 *v*/*v*) for 3 days and then cleared in saturated chloral hydrate as described previously [46]. H_2_O_2_ accumulation was stained with diaminobenzidine (DAB) for microscopic observation (Amresco, Solon, OH, USA). The presence of a substomatal vesicle was defined as an established infection unit. The infection sites and lengths of infection hyphae were stained with wheat germ agglutinin (WGA) conjugated to Alexa-488 (Invitrogen, Carlsbad, CA, USA) and observed under blue light excitation (excitation wavelength 450–480 nm, emission wavelength 515 nm). All microscopic examinations were conducted with an Olympus BX-53 microscope (Olympus Corporation, Tokyo, Japan). The H_2_O_2_ accumulation, hyphae lengths, and hyphae infection area (indicating the ability of fungal expansion) were calculated by DP-BSW software (Olympus) connected to the microscope. Fifty infection sites from three randomly selected leaf segments were examined for each treatment.

### 2.9. Protein Extraction and Western Blot

For the Western blot analysis, the total protein was extracted with protein extraction kits (Solarbio, Beijing, China) following the manufacturer’s instructions. Protein was separated by 10% SDS-PAGE. Gels were blotted onto a PVDF membrane (Merck Millipore, Burlington, MA, USA) using a transfer buffer at 64 V for 2 h. The procedure of Western blot analysis was described previously [47]. The membranes were washed three times with Tris-buffered saline (TBS, 5 min each) and then blocked with 5% nonfat dry milk in TBST (TBS with 0.1% Tween 20) for 1 h at 25 °C with 50 rpm shaking. Then, the membranes were incubated with primary antibody (mouse anti-GFP antibody (1:10,000, (Proteintech, IL, USA)), mouse anti-Bax antibody (1:10,000, Proteintech)) at 4 °C overnight. After washing three for 5 min in TBST, membranes were then incubated with goat anti-mouse IgG (H+L) antibody (Proteintech) as secondary antibody at 1:10,000 dilutions for 2 h, followed by three washes at room temperature. The Protein bands were detected with chemiluminescence horseradish peroxidase substrate (Merck Millipore, Burlington, MA, USA) and photographed following the manufacturer’s instructions.

## 3. Results

### 3.1. Complete Sequence and Organization of PsMV2 Genome

The full-length cDNA sequence of Puccinia striiformis mitovirus 2 (PsMV2) was 2658 nt in length, with a G+C content of 47.7%, which is similar to the mitochondrial genomes of fungi and plants, and shares characteristics with members of the genus *Unuamitovirus*. The cDNA sequence of PsMV2 was deposited in GenBank under the accession number OQ263177. It contains a single ORF being 2349 nt long that initiated at the nucleotide position 226 and terminated at position 2574, potentially encoding 783 amino acids (aa) residues with a calculated molecular weight of 88.24 kDa between the initiation AUG triplet and the termination UAA triplet (Figure 1A). In addition, the genome of PsMV2 had a 5′-untranslated region (UTR) of 225 nt and a 3′-UTR of 84 nt (Figure 1A). As with the rest of mitoviruses, the genome of PsMV2 presents a high percentage of A+U (52.3%) and contains a number of UGA and UGG codons that encode the amino acid tryptophan rather than acting as stop codons as in the universal genetic code. The ORF contains multiple codons (43 UGA and 47 UGG, UGA/UGG ratio < 0). The stem-loop structures of the 5′ and 3′ UTRs of PsMV2 were predicted using the Mfold program (http://rna.tbi.univie.ac.at//cgi-bin/RNAWebSuite/RNAfold.cgi, accessed on 12 October 2022), with initial ΔG values of −11.9 kcal/mol and −10.0 kcal/mol, respectively (Figure 1B). Additionally, the sequences of the 5′ and 3′ UTRs are inverted complementary, and have a potentially stable panhandle structure with a G value of −21.0 kcal/mol (Figure 1B).

### 3.2. Phylogenetic Analysis of the PsMV2

Based on the conserved domain database (CDD) search on the ORF-encoded RNA-dependent RNA polymerase (RdRp), the conserved RdRp domain belongs to the family *Mitoviridae* (RdRp: 180–466 aa). Multiple sequence alignment revealed that the RdRp of PsMV2 contains six conserved motifs (I–VI) (Figure 2A), which is a specific characteristic of the genus *Unuamitovirus* [48]. A homology search with BLASTP showed that this 88.24 kDa protein was most closely related to the RNA-dependent RNA polymerases (RdRps) of Cronartium ribicola mitovirus 1, Puccinia striiformis mitovirus 1 and Helicobasidium mompa mitovirus 1–18 (Appendix A). The virus identified in the *Pst* strain GS-1 (race CYR31) exhibited the highest similarity to Cronartium ribicola mitovirus 1 with 48% identity. The species demarcation criteria in the genus *Mitovirus*, defined by the ICTV 9th Report (https://talk.ictvonline.org/, accessed on 12 January 2023), indicate that mitoviruses with homologies in the amino acid sequence of RdRp proteins greater than 90% belong to different strains of the same mitovirus species. Therefore, we can conclude that the PsMV2 in the *Pst* strain GS-1 (race CYR31) corresponds to a novel mycovirus, a tentative member of a new species, belonging to the genus *Unuamitovirus* of mitochondrial mycoviruses with +ssRNA genomes. We have named this novel mycovirus, the first reported in this fungal pathogen, Puccinia striiformis mitovirus 2 (PsMV2). To establish the relationship between PsMV2 and other mycoviruses, a Maximum likelihood (ML) phylogenetic tree was constructed based on the RdRp aa sequence of PsMV2 and other related viral sequences, including members of the family *Mitoviridae* and representative members of the family *Narnaviridae* (Figure 2B). In the obtained phylogenetic tree, PsMV2 was grouped with unuamitoviruses, such as the Cronartium ribicola mitovirus 1 (Figure 2B). Therefore, we concluded that PsMV2 was a new member of the recently proposed genus of *Unuamitovirus* within the family *Mitoviridae* [16].

### 3.3. Transcript Level of PsMV2 Is Upregulated during the Early Infection Stage

To investigate the transcript level of PsMV2 during *Pst* infection stages, quantitative RT-PCR (qRT-PCR) was used. Results indicated that the transcription level of PsMV2 increased approximately 5.4–9 fold compared to urediospores during the penetration stages of *Pst* strain GS-1 (race CYR31) on wheat, 6–24 hpi (Figure 3). However, it decreased to about 1.6–5 fold during 48–96 hpi and was barely detectable at 120–256 hpi (Figure 3). Therefore, it was found that the transcription level of PsMV2 increased during the early infection stages of the host fungus, especially during haustorial and hyphal formation, and decreases drastically during the late “parasitic/biotrophic” stage of infection and sporulation.

### 3.4. PsMV2 Suppressed Bax-Induced Cell Death

Because *Pst* is not flexible for transformation, we used the transient expression method mediated by *A. tumefaciens* to verify the ability of PsMV2 to suppress Bax-triggered PCD, which is similar to the defense-related HR in plant cells [49,50]. One criterion for the suppression of plant immunocompetence is the inhibition of programmed cell death [51]. *PstGSRE1* which could suppress Bax-triggered PCD in *N. benthamiana* was used as a positive control [43]. For the negative control (only GFP), they did not abolish the function of Bax (Figure 4A). When chlorophyll was removed from tobacco leaves, macroscopic cell death was observed in the negative zone (Figure 4A). By contrast, we observed that PsMV2 could also suppress Bax-triggered PCD (Figure 4A). The protein expression of PsMV2-GFP, GFP and Bax were detected by Western blot with an anti-GFP antibody/anti-Bax antibody (Figure 4B), which indicated that PsMV2 was responsible for the suppression of PCD induced by Bax.

### 3.5. Silencing of PsMV2 Significantly Reduces the Virulence of Pst

To determine whether PsMV2 is involved in the pathogenicity of *Pst* infection in wheat, barley stripe mosaic virus (BSMV)-mediated HIGS was used to silence the expression of PsMV2 in *Pst* during infection. For the BSMV-HIGS assay, two cDNA fragments of the PsMV2 gene were silenced individually, the constructs were designated BSMV: PsMV2-1as and BSMV: PsMV2-2as, respectively (Figure 5A). At 12 days after BSMV inoculation, all of the Su11 wheat infected with BSMV: γ, BSMV: PsMV2-1/2as expressed similar phenotypes of mild chlorotic mosaic symptoms on the fourth leaf. Su11 seedlings inoculated with BSMV: TaPDS exhibited strong photobleaching symptoms, confirming that the BSMV-mediated gene silencing system functioned correctly and could be used in further experiments (Figure 5B). Subsequently, the fourth leaf of wheat was inoculated with fresh ungerminated urediospores of CYR31 (GS-1), and rust phenotypes were photographed at 14 dpi. After *Pst* inoculation, the number of rust pustules was significantly reduced in the wheat leaves inoculated with BSMV: PsMV2-1/2as compared with the control plants (Figure 5C). Total genomic DNA was isolated from wheat leaves infected with *Pst*, and the relative levels of *PsEF1* and *TaEF1* were quantified by qRT-PCR. At 10 days after inoculation, relative to the control plants, the biomass ratio of *Pst* in leaves treated with BSMV: PsMV2-1as and BSMV: PsMV2-2as were significantly reduced by 36% and 57%, respectively (Figure 5E). To test the silencing efficiency of BSMV-HIGS, qRT-PCR was used to assay the relative transcript level of PsMV2 at 24, 48, and 120 hpi with *Pst* race CYR31 (strain GS-1). Compared with control plants, the transcript level of BSMV: PsMV2-1as-infected leaves were reduced by 83%, 80% and 45% at 24, 48, and 120 hpi, respectively (Figure 5D). Similarly, in BSMV: PsMV2-2as-infected leaves, the transcript level of PsMV2 was reduced by 80%, 41%, and 35% at 24, 48, and 120 hpi, respectively (Figure 5D). These results indicated that the expression of PsMV2 was partially knocked down by HIGS, and silencing of PsMV2 reduced the pathogenicity and restricted the growth and development of *Pst* on wheat plants.

### 3.6. HIGS of PsMV2 Inhibits the Growth of Pst Mycelia and Increases the Accumulation of H_2_O_2_ in Wheat Cells

To evaluate the histological responses associated with PsMV2 silencing, we analyzed hyphal length and infection areas at the infection site. At 24, 48, and 120 h after inoculation with *Pst* race CYR31 (strain GS-1), the histological changes in the pathogen were observed by microscopy and were estimated by DP-BSW software (Figure 6A). The fluorescence microscope revealed that the hypha length and infected area were significantly less in BSMV: PsMV2 knock-down plants than in BSMV: γ-infected plants at 48 and 120 hpi (Figure 6B,C).

The accumulation of reactive oxygen species and the occurrence of HR occur in the resistance of wheat in response to rust fungus [46]. To analyze the host response, we analyzed H_2_O_2_ accumulation upon inoculation of wheat with *Pst* race CYR31 (strain GS-1) (Figure 7A). The results showed that H_2_O_2_ accumulation was expressively increased after silencing treated BSMV: PsMV2-1/2as leaves at 24 and 48 hpi (Figure 7B).

### 3.7. Wide Distribution of PsMV2 in Wheat Stripe Rust

To detect the host range of the PsMV2, we performed RT-PCR for a range of plant and fungal samples (Figure 8). PsMV2 was detected in a wide range of *Pst* isolates differing in race and geographic location (Appendix A), as well as in *P. graminis* f. sp. *tritici* and *P. recondita* f. sp. *tritici*. On the other hand, PsMV2 was not detected in plant samples, including wheat, *Mahonia fortune* (Lindl.) Fedde, *Berberis thunbergii* DC, *Hordeum vulgare* L, and *Brachypodium sylvaticum* (Huds.) Beauv (Figure 8). These results indicate that PsMV2 might be a parasitic mycovirus widespread in wheat stripe rust fungus.

## 4. Discussion

In recent years, with the development of next-generation sequencing technologies, an increasing number of mycoviruses have been identified [4,16,52]. Studying mycovirus pathogenicity is important for a better understanding of the molecular biology of viruses [26,38,53]. In this study, we described the complete nucleotide sequence, genome organization and biological characteristics of a novel mycovirus PsMV2 in *Pst*, the causal agent of wheat stripe rust. Phylogenetic analyses of RdRp strongly supported the conclusion that PsMV2 represents a new member of the genus *Unuamitovirus* within the family *Mitoviridae*. In addition, PsMV2 can suppress programmed cell death (PCD) triggered by Bax. Using host-induced gene silencing (HIGS) to knockdown the expression of RdRp in PsMV2 leads to decreased virulence of *Pst*. These results demonstrate the positive effect of PsMV2 on *Pst* virulence. Moreover, we also found that PsMV2 is widely parasitic in rust fungus. This research expands our knowledge of the ecology and evolution of viruses in the family *Mitoviridae* and offers insights into the sustainable control of wheat rust disease.

Viruses of the family *Mitoviridae* are +ssRNA viruses, encoding an RdRp protein, and the viruses exist as RNA-RdRp nucleoprotein complexes [14]. Mitoviruses essentially replicate within the mitochondria of their hosts [20]. Consequently, as with many fungal and plant mitochondrial genomes, mitoviruses share A-U richness [14]. In the mitovirus genome, the UGA codon encodes the aa tryptophan, not a stop codon as it does in the universal genetic code [20]. Our results show that the genome of PsMV2 does have a high proportion of A+U (52.3%) (Figure 1A). Another feature of mitovirus genomic RNA is the presence of stem-loop structures at both the 5′ end and the 3′ end, due to the presence of inverted complementary sequences at their termini [48], which is the situation for many mitoviruses found in various fungi, such as *Fusarium oxysporum* f. sp. dianthi mitovirus 1, Sclerotinia sclerotiorum mitovirus 1, and Melanconiella theae mitovirus 1 [17,22,26]. These structures may play an important role in a number of aspects of mitovirus replication, including the protection of naked ssRNA from degradation and as a recognition site for RdRp [48,54]. PsMV2 could be folded into potentially stable stem-loop structures (Figure 1B). The long-term co-evolution of mitoviruses and their fungal hosts would lead to important sequence differences in the mitovirus genome [48]. According to the phylogenetic tree, some mitoviruses isolated from different fungal species (e.g., PsMV2 and CrMV1) were relatively closely related in terms of phylogeny. The different mitoviruses may have evolved to exist prior to the differentiation of their fungal hosts.

Studying mycovirus pathogenicity is important for a better understanding of the molecular biology of viruses [26,38,53]. Mycoviruses have different effects on fungi, ranging from inducing a cryptic state to increasing the capacity of the host to produce disease (hypervirulence) [27]. Although previously reported mitoviruses have little or no effects on their natural hosts, some mitoviruses are associated with hypovirulence to fungal hosts, making them a potential alternative method of biological control of plant diseases [55]. Pst is an obligate biotrophic fungus without a protoplast transformation system, making it difficult to obtain sufficient amounts of experimental materials from obligate host fungi, and it can cause asymptomatic infection. In fungi, RNA silencing (or RNA interference) is thought to serve primarily as a defense mechanism against invasive nucleic acids and viruses [56]. Antiviral RNA silencing is substantiated in fungi, e.g., *Fusarium graminearum*, *Cryphonectria parasitica* and *Pyricularia oryzae* [57,58,59]. In the present study, we used a BSMV-HIGS approach to determine the role of PsMV2 in the wheat-*Pst* interaction. In the normal condition, it was found that PsMV2 infecting *Pst* race CYR31 (strain GS-1) underwent significant proliferation during the initial infection of wheat seedlings (Figure 3). On the other hand, when *Pst* race CYR31 (strain GS-1) invaded wheat seedlings in which RNA silencing against PsMV2 is activated using HIGS, it was revealed that the amount of PsMV2 infecting *Pst* decreased, and subsequently, the disease symptoms, fungal biomass, and mycelial dispersal by *Pst* decreased compared to the normal condition (Figure 5). These results suggested that PsMV2 is essential for the growth and development of *Pst*, which may be related to insufficient carbon and energy supply (Figure 6). In addition, we found that PsMV2 might be a parasitic mycovirus widespread in field *Pst* isolates (Figure 8 and Appendix A) it is, therefore, possible that the gain of PsMV2 confers selective advantage (e.g., hypervirulence) and has been stabilized by *Pst* during evolution.

In summary, PsMV2 is a novel +ssRNA mitovirus identified from the fungus *Pst*. PsMV2 plays a positive role in the virulence of *Pst*. The detection of PsMV2 in wheat rust fungus obtained from 10 different countries indicates that PsMV2 has a wide geographical distribution in *Pst*. Future research will be focused on generating PsMV2-RNAi transgenic material and exploring whether it has broad-spectrum disease resistance, thus providing a new strategy against rust pathogens in wheat.

## Figures and Tables

**Figure 1 viruses-15-01265-f001:**
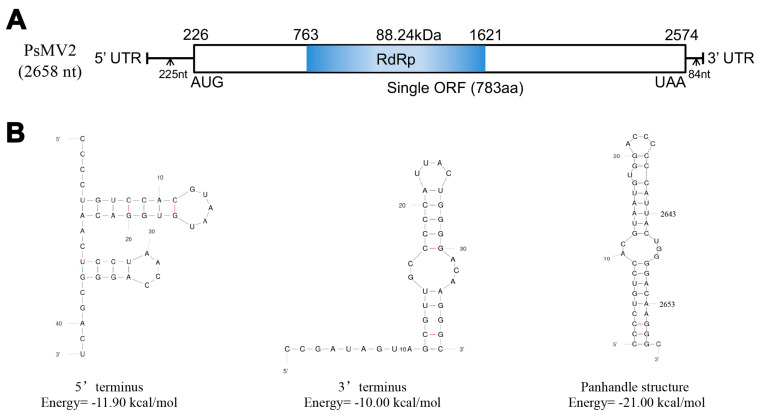
The genome organization of Puccinia striiformis mitovirus 2 (PsMV2). (**A**) Schematic representation of PsMV2 RNA genome, with key nucleotides and amino acids indicated. The open reading frame (ORF) and the untranslated regions (UTRs) are indicated by a hollow rectangle and single black lines, respectively. The conserved domain of RNA-dependent RNA polymerase (RdRp) is indicated by a blue bar. (**B**) Predicted secondary structures of the 5′ (left) and 3′ termini (middle) of PsMV2. A putative panhandle structure (right) formed by the inverted complementarity at the terminal sequences is also shown.

**Figure 2 viruses-15-01265-f002:**
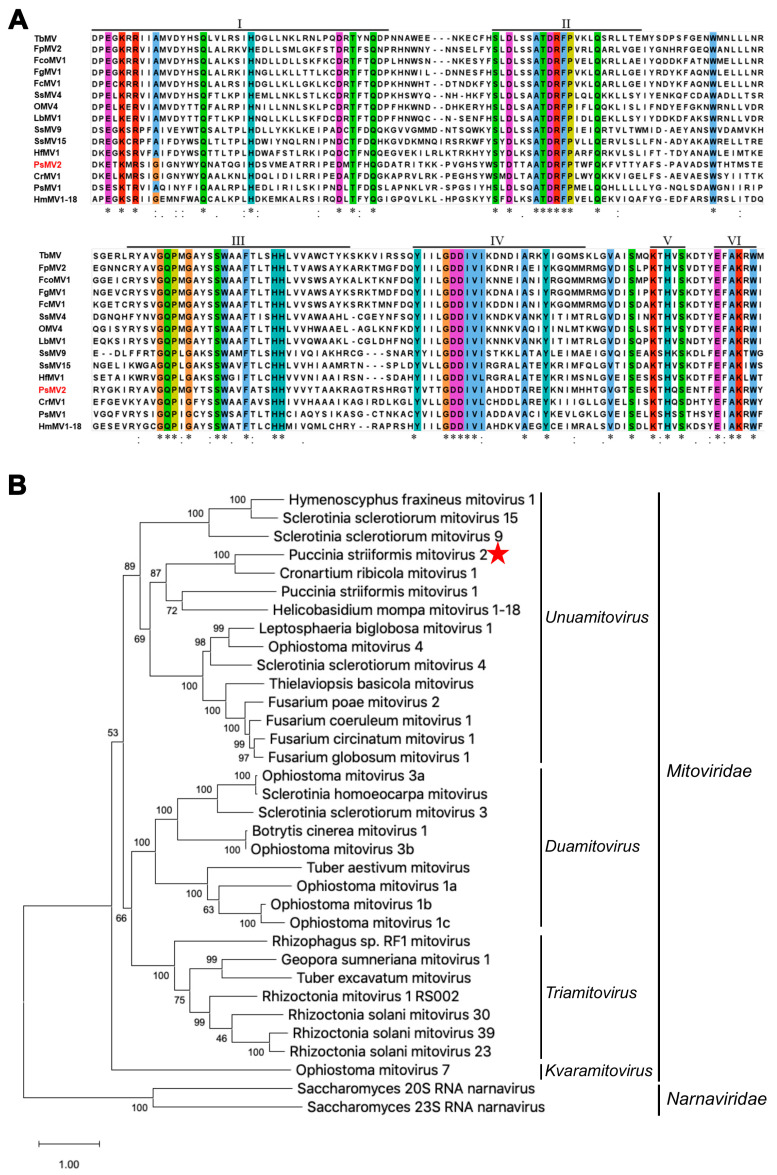
Phylogeny of Puccinia striiformis mitovirus 2 (PsMV2). (**A**) Multiple alignment of PsMV2 RdRp motifs with those of selected viruses of the genus *Unuamitovirus* using the ClustalX program. Six conserved motifs characteristic of RdRps of unuamitoviruses are indicated by lines above the sequences, and their positions are indicated by Roman numerals (I–VI). Shaded areas indicate identical amino acid residues. Asterisks, colons and dots indicate identical amino acid residues, conservative variations and semi-conservative variations, respectively. (**B**) Phylogenetic analyses of the RdRp domains of PsMV2, other mitoviruses and narnaviruses, based on the Maximum likelihood (ML) method and JTT model with bootstrapping analysis of 1000 replicates in MEGA 11.0. See Appendix A for abbreviations of virus names and viral protein accession numbers used for phylogenetic analysis.

**Figure 3 viruses-15-01265-f003:**
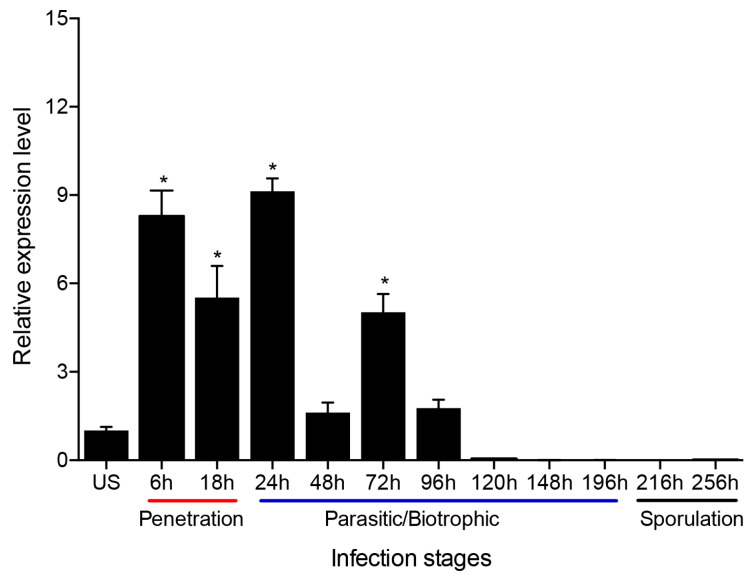
Relative transcript levels of PsMV2 at different *Pst* infection stages. Wheat leaves (Su11) inoculated with fresh urediospores (US, CYR31 (GS-1)) were sampled at different time points according to the infection stage of *Pst*. Urediospores were used as a control. Relative transcript level of PsMV2 was calculated using the comparative 2^−ΔΔCT^ method. *PstEF1*-F/R was used as the reference gene. Mean and standard deviation were calculated with data from three independent biological replicates. Differences were assessed using Dunnett’s test, and asterisks indicate *p* < 0.05.

**Figure 4 viruses-15-01265-f004:**
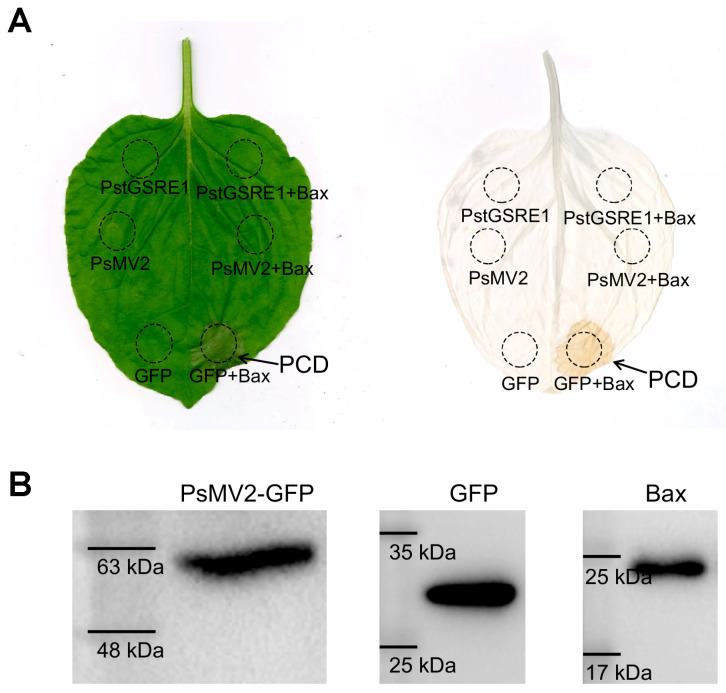
PsMV2 suppresses Bax-mediated cell death in *N. benthamiana*. MgCl_2_ buffer or *A. tumefaciens* cells carrying the PsMV2-GFP, PstGSRE1 or GFP vector were infiltrated into the leaf of *N. benthamiana* within the regions indicated by the dashed lines, followed after 24 h by either no further infiltration (left side) or infiltration with *A. tumefaciens* cells carrying the Bax (right side). The phenotype of cell death was scored and photographed at 5 days after infiltration with PstGSRE1. The same leaf was examined before (left) and after (right) being decolorized with the destaining solution. (**B**) Western blot analysis of GFP (Predicted molecular mass of 26 kDa), Bax (Predicted molecular mass of 21 kDa), PsMV2-GFP (Predicted molecular mass of 57.83 kDa (31.83 kDa + 26 kDa)) in total protein extracts from *N. benthamiana* leaves. Detection of protein expression was conducted by immuno-detection with anti-Bax or anti-GFP antibodies. Proteins were extracted from the leaves of *N. benthamiana* co-transfected with the constructs indicated in (**A**).

**Figure 5 viruses-15-01265-f005:**
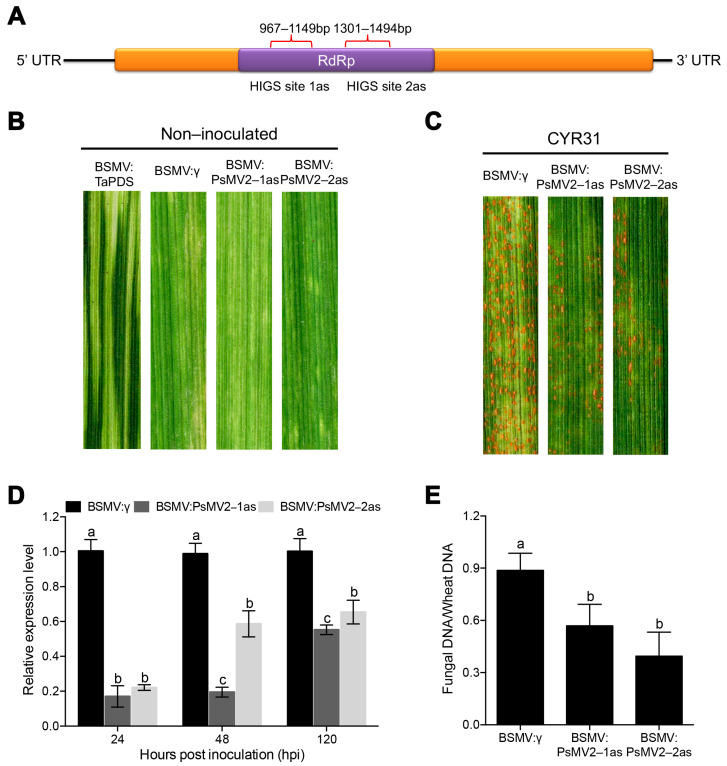
Functional evaluation by BSMV-mediated HIGS on the role of PsMV2 in *Pst* pathogenicity. (**A**) The indicated specific sequence regions were selected for BSMV-mediated transient silencing. HIGS site 1as: PsMV2-1as; HIGS site 2as: PsMV2-2as. (**B**) The leaves of BSMV: γ, BSMV: TaPDS-as, BSMV: PsMV2-1/2as showed mild chlorotic mosaic symptoms 12 days after inoculation (dpi). Mock, wheat leaves treated with 1× FES buffer. (**C**) Seedlings of cultivar Suwon 11 were inoculated with the labeled BSMV constructs on the second leaf for 14 days and then inoculated with urediospores of *Pst* race CYR31 (strain GS-1) on the fourth leaf. (**D**) Relative transcript levels of PsMV2 in leaves inoculated with *Pst* were assayed by qRT-PCR at 24, 48 and 120 h after inoculation with *Pst* race CYR31 (strain GS-1). Data were normalized to the transcript level of *PstEF1*. (**E**) qPCR measurement of fungal biomass. Ratio of fungal to wheat nuclear content was used for fungal *PstEF1* and wheat *TaEF1* genes, respectively. Genomic DNA was extracted from the fourth leaf from three different plants at 10 dpi. Values represent the means ± SE of three independent samples. Different letters (a, b and c) in (**D**,**E**) denote significant differences (*p* < 0.05 determined by Tukey’s post hoc test).

**Figure 6 viruses-15-01265-f006:**
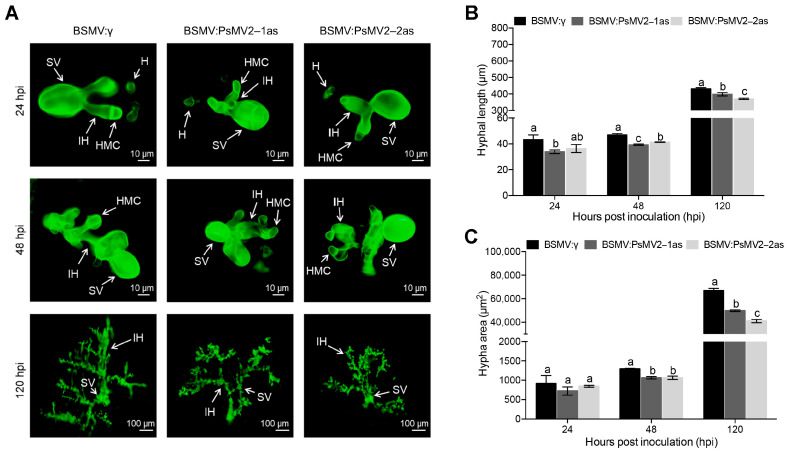
Histological changes in *Pst* race CYR31 (strain GS-1) growth of PsMV2-silenced plants. (**A**) The fungal structures were stained with wheat germ agglutinin in wheat leaves inoculated with BSMV: γ, BSMV: PsMV2-1/2as at 24, 48 and 120 hpi, and were observed under a fluorescence microscope. HMC: haustoria mother cell; IH: infectious hyphae; H: Haustorium; SV: substomatal vesicle. (**B**,**C**) Hyphal lengths (**B**) and infection areas (**C**) in BSMV-infected plants inoculated with the CYR31 (GS-1) isolate at 24, 48 and 120 hpi were calculated by DP-BSW software. All results were obtained from 50 infection sites and three biological replications were performed. Values represent the means ± SE of three independent samples. Different letters (a, b and c) in (**B**,**C**) denote significant differences (*p* < 0.05 determined by Tukey’s post hoc test).

**Figure 7 viruses-15-01265-f007:**
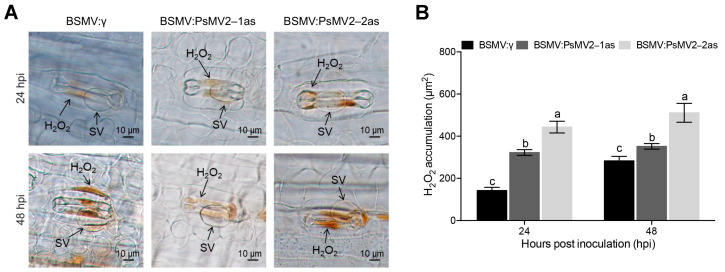
Observations of H_2_O_2_ accumulation in PsMV2-silenced plants that were inoculated with *Pst* race CYR31 (strain GS-1). (**A**) H_2_O_2_ accumulation was observed at infection sites by microscopy. Scale bars, 10 µm. SV, substomatal vesicle. (**B**) The accumulation of H_2_O_2_ was measured by calculating the DAB-stained area at each infection site using the DP-BSW software at 24 and 48 hpi. All results were obtained from 50 infection sites and three biological replications were performed. Values represent the means ± SE of three independent samples. Different letters (a, b and c) in (**B**) denote significant differences (*p* < 0.05 determined by Tukey’s post hoc test).

**Figure 8 viruses-15-01265-f008:**
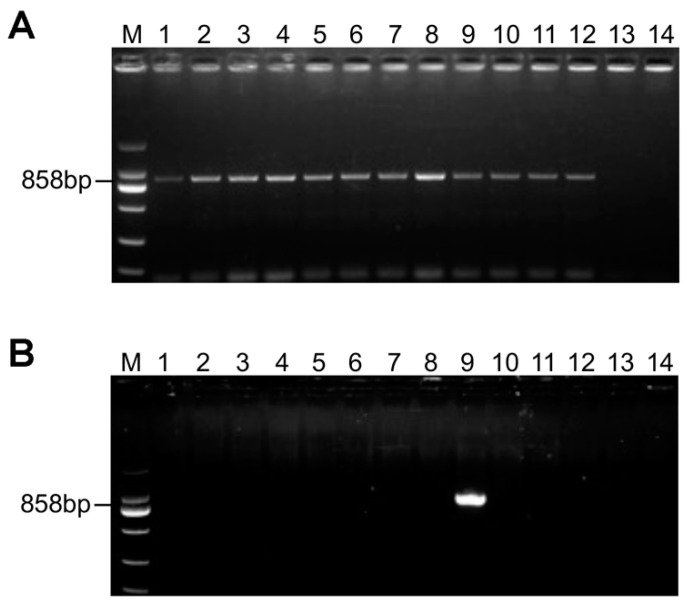
RT-PCR detection of PsMV2 in *Pst* isolates and other wheat. (**A**) Isolates representing same *Pst* races were analyzed by genomic PCR for the presence of PsMV2. 1: CYR23 (*Pst* race CYR23); 2: CYR31 (*Pst* race CYR31); 3: CYR32 (*Pst* race CYR32); 4: CYR35 (*Pst* race CYR35); 5: CYR34 (*Pst* race CYR34); 6: CYR29 (*Pst* race CYR29); 7: CYR20 (*Pst* race CYR20); 8: CYR22 (*Pst* race CYR22); 9: V26 (*Pst* race V26). 10: *Puccinia graminis* f. sp. *tritici*; 11: *P. recondita* f. sp. *tritici*; 12: PST-BH; 13: *P. sorghi*; 14: ddH_2_O; (**B**) PCR detection of PsMV2 in different wheat varieties. 14: ddH_2_O; 1–8: wheat varieties (Su11, Mingxian 169, Xiaoyan 22, Nongke 199, Nongke 9204, Jimai 22, Fielder, Kangyin 655); 9: Positive control (*Pst* race CYR31); 10: *Mahonia fortune* (Lindl.) Fedde; 11: *Berberis thunbergii* DC; 12: *Hordeum vulgare* L; 13: *Brachypodium sylvaticum* (Huds.) Beauv; 14: ddH_2_O. Lane Marker, DL2000 DNA ladder marker.

## Data Availability

The whole PsMV2 genome sequences havebeen deposited in GenBank under accession numbers OQ26317. Other data are available in Appendix A.

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
