# Peer review of "A Novel Mitovirus PsMV2 Facilitates the Virulence of Wheat Stripe Rust Fungus"

_viruses, 2023, doi:10.3390/v15061265_

Round 1
Reviewer 1 Report
The authors found a new mitovirus from wheat stripe rust fungus, determined its genome, and showed that it is a new member of the genus Unuamitovirus by molecular phylogenetic analysis. Furthermore, they revealed the dynamics of this mitovirus during the infection of wheat by the host fungus and suggested that it helps the host fungus infect wheat by suppressing plant programmed cell death. This study revealed very interesting characteristics of mitoviruses infecting obligate biotrophic fungi, which have been poorly studied, and is expected to drive future research on mitoviruses. It will therefore be of great interest to researchers in this field.
I have the following concerns.
1. "Pst race CYR31" is mentioned frequently, but the strain name should be specified. Is Pst GS-1 the same as race CYR31? (Please add "race" in Line 84.) Please provide a detailed description of how Pst GS-1 was isolated (Is Strain GS-1 a single-spore isolated strain?)
2. Please confirm whether using t-tests for multiple comparisons is appropriate for statistical processing. If a parametric analysis is possible, I think Dunnett's test should be used for Figure 3 and the Turkey-Kramer test should be used for Figures 5, 6, and 7.
3. Regarding molecular phylogenetic analysis, according to the proposal to the ICTV for the classification of the Mitoviridae family, the study group used the ML method for the analysis. Therefore, please estimate the phylogenetic tree (Figure 2B) using the ML method instead of the NJ method, and please also describe the model selected for the analysis.
4. In the Results section, only the obtained results should be reported in the past tense, while any inferences drawn from them should be discussed in the Discussion section. Please arrange the figures in numerical order.
5. Regarding the determination of the draft genome of PsMV2, please provide more detailed information since it is not described in Result 3.1. I also think it is necessary to describe the UGA/UGG ratio of PsMV2.
6. Figure 3: Is it known how the dynamics of host fungal mitochondria change during wheat infection? Since the number of host mitochondria is altered, does it not seem that PsMV2 is also changing as a result? In section 3.3, please provide a detailed description of the results as the data presented in Figure 3 is insufficiently described.
7. Figure 4: How was Bax expressed? Why was the PVX vector, which is used for VIGS, used? Did the PVX spread with replication outside the infiltration site? Were the controls GFP and PstGSRE1 also inserted into the same PVX vector? Was pGR106 constructed by the authors themselves, or was it a gift from other researchers? Where did the authors obtain the GV3101 strain? Please provide more detailed information in the Materials and Methods section. Were the protein accumulation levels of PstGSRE1, PsMV2-GFP, GFP, and Bax confirmed? Can the authors show a fluorescent image of GFP in this experiment?
8. Figure 5: Is the method of knocking down genes of pathogenic fungi infecting plants using HIGS commonly used? Is there previous knowledge that this method is applicable to viruses localized in mitochondria? If this is the first time it was demonstrated with a mitovirus, the authors should emphasize and describe it in the discussion. If this is the first time, as a reader, I would like to see a control where the knockdown of a gene of Pst (possibly in mitochondria) is confirmed, in addition to using wheat PDS as a control, which I think is not sufficient. How did the authors distinguish between transcripts (mRNA) and genomic RNA? What was the sequence homology between the two cDNA fragments (Line 299; site1as and site 2as) and the host (including mitochondria) genome?
9. The Discussion section should focus on the data obtained in this study, properly citing them. Please add a more detailed discussion of the data presented in each figure. Then, please develop a comprehensive discussion that the authors would like to make based on the results obtained in this study.
10. Isn't it correct to number each reference in the reference list and cite that number in the text when referring to the respective reference? Please check carefully the reference list. For example, the correct information in lines 441-442 is “Akata, I., Keskin, E., and Sahin, E. Molecular characterization of a new mitovirus hosted by the ectomycorrhizal fungus Albatrellopsis flettii. Arch. Virol. 2021, 166, 3449–3454. https://doi.org/10.1007/s00705-021-05250-4.” I cannot find Marzano and Domier, 2016 (Line 374) in the list.
11. Did the authors not find any Pst that were not infected with PsMV2 in this study? Have the authors investigated the transmission rate of PsMV2 using single-spore isolation or other methods? Why do the authors think this widespread virus was discovered for the first time in this study? Looking at the gel electrophoresis image in Figure 8, it seems that there were differences in the intensity of the bands depending on the sample. Did this mean that there is diversity in the PsMV2 sequence among the samples? Have the authors not analyzed the sequences of these viruses?
Here are some minor revisions.
Lines 18 and 262: “PsMV2 was highly expressed” should be “PsMV2 multiplied highly”. The expression “The virus was expressed.” feels strange to me.
Line 20: “HIGS” should be written out in full and not abbreviated here.
Lines 65-67: “While most … (Xu et al., 2015)”. If my memory serves me correctly, since the effects of most mitoviruses on their host fungi are not well understood, I recommend revising this sentence to the following. “While some mitoviruses can lead to hypovirulence of their fungal hosts (Xu et al., 2015), others have beneficial effects on the virulence of them.”
Lines 67-68: “For example, …2016)”. Please describe the mechanism by which the virus enhances the virulence of the host, as this is relevant to this study. Alternatively, it may be appropriate to discuss and contrast the mechanism of PsMV2 in the Discussion section.
Lines 68-69: “Mitoviruses … interactions.” Specifically, how? I would like the authors to describe specific examples of fundamental biology that can only be understood through the study of mitoviruses.
Line 78: “characterization” and “biology” should be “characterized” and “biological”, respectively.
Line 80: “evolutionary” would be “evolution”.
Line 99: “nuclease (Takara) so as to digest genomic DNA and ssRNA (Takara, Dalian, China),” should be “nuclease (Takara, Dalian, China) to digest genomic DNA and ssRNA,”.
Line 104: “RQ1 DNase I” should be “RQ1 RNase-Free DNase (Promega, City, Country)”.
Line 108: Please add the city and country information to “Illumina”.
Line 110, 138: “(MNI, K1622)” should be “(Thermo Fisher Scientific, City, Country)”.
Lines 111, 139: “(CWBIO, China) should be “(CWBIO, City, China)”.
Lines 122-123: Is it unnecessary to cite the literature (Sigrist et al., 2012) to the PROSITE database?
Lines 126-127: Is “Madden et al., 1996” the correct reference to be cited here?
Line 135: “a specific primer” should be “a specific primer pair”.
Line 137: “China, Beijing” should be “Beijing, China”.
Line 137: “were” should be “was”.
Lines 142-143: Please add company information (Name, City, Country) to “ChamQ SYBR qPCR Master Mix”.
Line 176: Please write down the composition of “1×FES buffer”.
Line 178: “leaves” should be “leaf”.
Lines 179-181: “When …16±2°C”. Is the following interpretation of this sentence correct? “When photobleaching symptoms appeared on leaves silenced with BSMV:TaPDS, the fourth leaves of seedlings were inoculated with fresh Pst race CYR31 urediospores on Suwon11 and maintained at 16±2°C.”
Lines 181-182: “For silencing … assay.” Is the following interpretation of this sentence correct? “To evaluate the silencing efficiency and biomass ratio, wheat transgenic lines were analyzed using RT-qPCR assay.”
Line 188 and Figure 5A: What criteria are used to distinguish the use of "VIGS" and "HIGS"? If there are no such criteria, please use "HIGS" consistently throughout the paper.
Line 210: “Tokio” should be “Tokyo”.
Line 221: “molecular weight” should be “molecular mass”.
Line 221-223: “as predicted…html” should be explained in the Materials and Methods section.
Figure 1A: Please confirm that the C-terminal residue of the amino acid in the genome map of PsMV2’s RdRp region should be “1621” instead of “1620”.
Lines 249-250: “Taken …Mitoviridae” should be moved to the Discussion section.
Line 273: Please modify “Urediospores” to “Urediospores (US)” to clarify what “US” represents in Figure 3.
Lines 284-285: Please provide specific details regarding “the function”.
Line 291: Did the authors use “GFP” or “eGFP” in this study?
Line 297: “wheat” should be “in wheat”.
Line 327: “were estimated by DP-BSW software” should be explained in detail in the Materials and Methods section.
Line 334-335: “These results… leaves.” should be moved to the Discussion section.
Line 340: Please add an explanation for what “H” represents in Figure 6A.
Line 347: “measured” should be “observed”.
Line 348: “by calculating … DP-BSW software” should be explained in detail in the Materials and Methods section.
Lines 364-366: Please include the strain name in the description of each sample (race) name.
Supplementary Table S3: Please add the information on race to the samples.
Line 411: I believe it is “some”, not “most”.
To properly convey great results and claims, it is recommended to use an English proofreading service.
Reviewer 2 Report
Lines 2-3, change ‘A novel mitovirus PsMV2 facilitate the pathogenicity of wheat stripe rust’ to ‘A novel mitovirus PsMV2 facilitate the viulence of wheat stripe rust fungus’
Line 22, change ‘range of field Pst isolates’ to ‘range of field isolates of Pst’
Line 23, Line 84, Line 361, change ‘wheat stripe rust’ to ‘wheat stripe rust fungus’
Line 24, change ‘wide distributed in’ to ‘wide distribution in’
Line 31, ‘23 families’? please confirm the correct number.
Line 58, change ‘with’ to ‘has’
Line 65, change ‘While most’ to ‘Most’
Line 73, change ‘rust mycoviruses’ to ‘mycoviruses in rust fungus’
Line 74, Line 105, Line 200, Line 315, Line 317, Line 318, Line 319, Line 419, change ‘and’ to ‘, and’
Line 78, change ‘we characterization the’ to ‘we characterized the’
Line 83, change ‘strains, plant materials and growth conditions’ to ‘Strains, Plant Materials and Growth Conditions’
Line 86, change ‘Then fresh’ to ‘Fresh’
Line 90, change ‘Then’ to ‘Then, ’
Line 92, change ‘and maintained’ to ‘, and maintained’
Line 99, change ‘Takara’ to ‘TaKaRa’
Line 102, change ‘RNA extraction, cDNA cloning and sequencing’ to ‘RNA Extraction, cDNA Cloning and Sequencing’
Line 131, Line 238, change ‘analysis’ to ‘Analysis’
Line 137, change ‘China, Beijing’ to ‘Beijing, China’
Line 144, change ‘primers listed’ to ‘primers are listed’
Line 159, change ‘can be used as’ to ‘is used as’
Line 163, change ‘for Barley Stripe Mosaic Virus’ to ‘for barley stripe mosaic virus’
Line 167, Line 169, ‘in vitro’ should be italicized.
Line 178, change ‘the leaves’ to ‘, the leaves’
Line 199, change ‘observation of Pst growth in wheat leaves’ to ‘Observation of Pst Growth in Wheat Leaves’
Line 210, ‘Tokio’ is ‘Tokyo’?
Line 213, change ‘sequence and organization of PsMV2 genome’ to ‘Sequence and Organization of PsMV2 Genome’
Line 241, change ‘members of the family Mitoviridae’ to ‘including members of the family Mitoviridae’
Line 242, delete ‘included’
Line 243, delete ‘which showed that’
Line 246, change ‘of the Unuamitovirus genus’ to ‘of the genus Unuamitovirus’
Line 262, change ‘PsMV2 is highly expressed at the early stage of Pst infection wheat’ to ‘PsMV2 Is Highly Expressed at the Early Stage of Pst Infection Wheat’
Line 263, change ‘infection stages’ to ‘infection stages, ’
Line 277, change ‘suppressed Bax-Induced cell death’ to ‘Suppressed Bax-Induced Cell Death’
Line 306, change ‘the fourth leaves of wheat plants’ to ‘the fourth leaf of wheat’
Line 312, change ‘compared’ to ‘relative’
Lines 322-323, change ‘of PsMV2 inhibits the growth of Pst mycelia and increases the accumulation of H2O2 in wheat cells’ to ‘of PsMV2 Inhibits the Growth of Pst Mycelia and Increases the Accumulation of H2O2 in Wheat Cells’
Line 354, change ‘distribution of PsMV2 in wheat stripe rust’ to ‘Distribution of PsMV2 in Wheat Stripe Rust Fungus’
Line 357, change ‘it was also detected’ to ‘as well as’
Line 359, change ‘plant samples’ to ‘plant samples,’
Line 377, change ‘PsMV2’ to ‘PsMV2 in Pst’
Lines 385-386, change ‘sustainable wheat rust disease control’ to ‘sustainable control of wheat rust disease’
Line 397, change ‘Melanconiella theae’ to ‘and Melanconiella theae’
Lines 413-414, change ‘Wheat stripe rust as obligate biotrophic fungus’ to ‘Pst as obligate biotrophic fungus’
Line 424, change ‘pathogenic’ to ‘virulence’
Lines 446-447, the title should not be italicized.
Line 477, Line 513, Line 521, Line 551, change ‘Proc. Natl. Acad. Sci.’ to ‘Proc. Natl. Acad. Sci. U.S.A.’
Line 479, please confirm the last name and first name of authors.
Line 480, change ‘in china’ to ‘in China’
Line 519, ‘MBio’ is ‘mBio’?
Line 552, change ‘Mitovirus’ to ‘mitovirus’
None.
Round 2
Reviewer 1 Report
I am glad to see the ms revised. I apologize for not conveying my intention clearly in some parts. Now, I have the following concerns.
1. I would like to request once again that the authors please write the strain name of race CYR31 used in each experiment. It is important to know if the same strain was used in each experiment, especially if the GS-1 strain, which was used for PsMV2 sequencing, was also used in each experiment. Without this information, it is difficult to confirm if the results are consistent across the experiments.
2. Did the authors confirm that the Pst strains used in this study were not infected with other viruses such as PsMV1, apart from PsMV2?
3. I appreciate that the authors included the results of the western blot, but please provide detailed descriptions of the protein extraction and western blot methods in the Materials and Methods section.
4. Line 153, “pGR106”: I also performed VIGS using PVX-PDS from the pGR106 vector about 20 years ago. At that time, we used it with a kind gift from Sir David Baulcombe (University of Cambridge).
https://www.plantsci.cam.ac.uk/research/davidbaulcombe/methods/vigs
However, is the pGR106 vector the authors used different from the one?
5. Regarding molecular phylogenetic analysis, please describe the best model selected for the analysis. If a phylogenetic tree is inferred using an inappropriate model, an incorrect tree will be inferred. Therefore, without information about which best model was used, it is difficult to determine whether the inferred phylogenetic tree is accurate.
6. The authors claim that PsMV2 is a new member of the family Mitoviridae (a novel mitovirus), but does it meet the new species demarcation criteria for the family Mitoviridae? If so, it should be mentioned in the manuscript. (Please provide the species demarcation criteria while indicating that it meets these criteria.)
7. Please arrange the figures in numerical order. Specifically, please place Figure 5 after Figure 4. Please also remove “(Figure 5A)” in line 164 and include the term “BSMV-mediated HIGS” in the legend of Figure 5.
8. Regarding the determination of the draft genome of PsMV2, please provide more detailed information since it is not described in Result 3.1. I mean please show the draft genome obtained using the method described in the Materials and Methods section (e.g. "A draft genome of length XXX was obtained from XX clones derived from XX"). It is unclear from the Materials and Methods section why both Illumina and Thermo kits were used. It would be helpful if the results section could provide more detailed information about the status of the draft genome obtained.
9. In section 3.3, I mean, for example, I suggest explaining the results using the following framework: “The copy number of PsMV2 increased approximately 5-9 fold compared in urediospores during the penetration stages of Pst on wheat, 6-24 hpi. However, it decreased to about 1-4 fold during 48-96 hpi and was barely detectable at 120-256 hpi. Therefore, it was found that the copy number of PsMV2 increased during the early infection stages of the host fungus, especially during haustorial and hyphal formation, and decreases drastically during the late“parasitic/biotrophic” stage of infection and sporulation.”
The new sentence in lines 273-274 should be in the Discussion section.
10. Lines 433-437. I would recommend writing it as follows, but please verify if the content is accurate: “In the normal condition, it was found that PsMV2 infecting Pst CYR31 underwent significant proliferation during the initial infection of wheat seedlings (Figure 3). On the other hand, when Pst CYR31 invaded wheat seedlings in which RNA silencing against PsMV2 is activated using HIGS, it was revealed that the amount of PsMV2 infecting Pst decreased, and subsequently, the disease symptoms, fungal biomass, and mycelial dispersal by Pst decreased compared to the normal condition (Figure 5). These results suggested that PsMV2 is essential for the growth and development of Pst, which may be related to insufficient carbon and energy supply (Figure 6).”
What do the authors think PsMV2 is doing to compensate for the insufficient carbon and energy supply?
Here are some minor revisions.
Line 46. “coviruses infection” should be “covirus infections”
Line 83. “GS-1” should be “GS-1 (race CYR31)”
Line 96. “TaKaRa” should be “TaKaRa, Dalian, China”
Line 97. Please delete “(TaKaRa, Dalian, China)”
Line 127, [43]. The literature cited is for MEGA6. Please replace it with a literature citation for MEGA11.
Lines 142 and 143. Is it correct as 10 “mM”? Is it a mistake and should it be 10 “μM”?
Lines 160, 179, 190, 192, 293, and so on. Please unify either “d” or “days”.
Lines 245-246. “neigh-bor joining (NJ)” should be “Maximum likelihood (ML)”
Lines 302-303. The unit “kDa” should be followed by a space.
Line 418. “are relatively closely related in terms of phylogeny” should be “were relatively closely related in terms of phylogeny (Figure 2B)”
Line 424. “same” should be “some”.
Due to time constraints, I am unable to make detailed corrections. I ask the authors to pay meticulous attention and carefully review the manuscript.
To properly convey great results and claims, it is strongly recommended to use an English proofreading service.
